# High Fruit and Vegetable Consumption and Moderate Fat Intake Are Associated with Higher Carotenoid Concentration in Human Plasma

**DOI:** 10.3390/antiox10030473

**Published:** 2021-03-17

**Authors:** María Marhuenda-Muñoz, José Fernando Rinaldi de Alvarenga, Álvaro Hernáez, Anna Tresserra-Rimbau, Miguel Ángel Martínez-González, Jordi Salas-Salvadó, Dolores Corella, Mireia Malcampo, José Alfredo Martínez, Ángel M. Alonso-Gómez, Julia Wärnberg, Jesús Vioque, Dora Romaguera, José López-Miranda, Ramón Estruch, Francisco J. Tinahones, José Lapetra, J. Lluís Serra-Majem, Aurora Bueno-Cavanillas, Josep A. Tur, Vicente Martín Sánchez, Xavier Pintó, Miguel Delgado-Rodríguez, Pilar Matía-Martín, Josep Vidal, Clotilde Vázquez, Lidia Daimiel, Emilio Ros, Mercè Serra-Mir, Zenaida Vázquez-Ruiz, Stephanie K. Nishi, Jose V. Sorlí, María Dolores Zomeño, María Angeles Zulet, Jessica Vaquero-Luna, Rosa Carabaño-Moral, Leyre Notario-Barandiaran, Marga Morey, Antonio García-Ríos, Ana M. Gómez-Pérez, José Manuel Santos-Lozano, Pilar Buil-Cosiales, Josep Basora, Olga Portolés, Helmut Schröder, Itziar Abete, Itziar Salaverria-Lete, Estefanía Toledo, Nancy Babio, Montse Fitó, Miriam Martínez-Huélamo, Rosa M Lamuela-Raventós

**Affiliations:** 1Centro de Investigación Biomédica en Red Fisiopatología de la Obesidad y la Nutrición (CIBEROBN), Instituto de Salud Carlos III, 28029 Madrid, Spain; mmarhuendam@ub.edu (M.M.-M.); alvaro.hernaez@fhi.no (Á.H.); annatresserra@ub.edu (A.T.-R.); mamartinez@unav.es (M.Á.M.-G.); jordi.salas@urv.cat (J.S.-S.); dolores.corella@uv.es (D.C.); jalfmtz@unav.es (J.A.M.); angelmago13@gmail.com (Á.M.A.-G.); jwarnberg@uma.es (J.W.); mariaadoracion.romaguera@ssib.es (D.R.); jlopezmir@uco.es (J.L.-M.); restruch@clinic.cat (R.E.); fjtinahones@hotmail.com (F.J.T.); joselapetra543@gmail.com (J.L.); lserra@dcc.ulpgc.es (J.L.S.-M.); pep.tur@uib.es (J.A.T.); xpinto@bellvitgehospital.cat (X.P.); clotilde.vazquez@fjd.es (C.V.); eros@clinic.cat (E.R.); zvazquez@unav.es (Z.V.-R.); stephanie.nishi@urv.cat (S.K.N.); jose.sorli@uv.es (J.V.S.); mazulet@unav.es (M.A.Z.); marga.morey@yahoo.es (M.M.); angarios2004@yahoo.es (A.G.-R.); anamgp86@gmail.com (A.M.G.-P.); jsantos11@us.es (J.M.S.-L.); pilarbuilc@gmail.com (P.B.-C.); jbasora@idiapjgol.org (J.B.); olga.portoles@uv.es (O.P.); iabetego@unav.es (I.A.); etoledo@unav.es (E.T.); nancy.babio@urv.cat (N.B.); mfito@imim.es (M.F.); 2Department of Nutrition, Food Science and Gastronomy, School of Pharmacy and Food Sciences and XaRTA, Institute of Nutrition and Food Safety (INSA-UB), University of Barcelona, 08921 Santa Coloma de Gramenet, Spain; mmartinezh8@gmail.com; 3Food Research Center (FoRC), Department of Food Science and Experimental Nutrition, School of Pharmaceutical Sciences, University of São Paulo, 05508-000 São Paulo, Brazil; zehfernando@gmail.com; 4Centre for Fertility and Health, Norwegian Institute of Public Health, 0473 Oslo, Norway; 5Blanquerna School of Health Sciences, Universitat Ramon Llull, 08025 Barcelona, Spain; mzomeno@imim.es; 6August Pi Sunyer Biomedical Research Center (IDIBAPS), 08036 Barcelona, Spain; 7Department of Preventive Medicine and Public Health, University of Navarra, IdiSNA, 31008 Pamplona, Spain; 8Department of Nutrition, Harvard T.H. Chan School of Public Health, Boston, MA 02115, USA; 9Universitat Rovira i Virgili, Departament de Bioquímica i Biotecnologia, Unitat de Nutrició, 43204 Reus, Spain; 10Nutrition Unit, University Hospital of Sant Joan de Reus, 43201 Reus, Spain; 11Institut d’Investigació Sanitària Pere Virgili (IISPV), 43201 Reus, Spain; 12Department of Preventive Medicine, University of Valencia, 46010 Valencia, Spain; 13Unit of Cardiovascular Risk and Nutrition, Institut Hospital del Mar de Investigaciones Médicas (IMIM), 08007 Barcelona, Spain; mireiamalcampo@gmail.com (M.M.); HSchoeder@imim.es (H.S.); 14Center for Nutrition Research, Department of Nutrition, Food Sciences, and Physiology, University of Navarra, 31008 Pamplona, Spain; 15Precision Nutrition Program, IMDEA Food, CEI UAM + CSIC, 28049 Madrid, Spain; vicente.martin@unileon.es (V.M.S.); mdelgado@ujaen.es (M.D.-R.); lidia.daimiel@imdea.org (L.D.); 16Bioaraba Health Research Institute, Cardiovascular, Respiratory and Metabolic Area, 01009 Vitoria-Gasteiz, Spain; luna_jess_@hotmail.com (J.V.-L.); itziar_salaverria@yahoo.es (I.S.-L.); 17Osakidetza Basque Health Service, Araba University Hospital, University of the Basque Country UPV/EHU, 01009 Vitoria-Gasteiz, Spain; 18Department of Nursing, School of Health Sciences, Instituto de Investigación Biomédica de Málaga (IBIMA), University of Málaga, 29010 Málaga, Spain; abueno@ugr.es; 19CIBER de Epidemiología y Salud Pública (CIBERESP), Instituto de Salud Carlos III, 28029 Madrid, Spain; vioque@umh.es (J.V.); lnotario@umh.es (L.N.-B.); 20Unit of Nutritional Epidemiology, Miguel Hernandez University, ISABIAL-FISABIO, 03010 Alicante, Spain; 21Health Research Institute of the Balearic Islands (IdISBa), 07120 Palma de Mallorca, Spain; 22Department of Internal Medicine, Maimonides Biomedical Research Institute of Cordoba (IMIBIC), Reina Sofia University Hospital, University of Cordoba, 14004 Cordoba, Spain; 23Internal Medicine Service, Hospital Clínic, University of Barcelona, 08036 Barcelona, Spain; 24Department of Endocrinology, Virgen de la Victoria Hospital, Instituto de Investigación Biomédica de Málaga (IBIMA), University of Málaga, 29010 Málaga, Spain; 25Research Unit, Department of Family Medicine, Distrito Sanitario Atención Primaria Sevilla, 41010 Sevilla, Spain; 26Research Institute of Biomedical and Health Sciences (IUIBS), University of Las Palmas de Gran Canaria & Centro Hospitalario Universitario Insular Materno Infantil (CHUIMI), Canarian Health Service, 35016 Las Palmas de Gran Canaria, Spain; 27Department of Preventive Medicine and Public Health, University of Granada, 18016 Granada, Spain; 28Research Group on Community Nutrition & Oxidative Stress, IUNICS, University of Balearic Islands, 07122 Palma de Mallorca, Spain; 29Institute of Biomedicine (IBIOMED), University of León, 24071 León, Spain; 30Lipids and Vascular Risk Unit, Internal Medicine, Hospital Universitario de Bellvitge, Hospitalet de Llobregat, 08908 Barcelona, Spain; 31Division of Preventive Medicine, Faculty of Medicine, University of Jaén, 23071 Jaén, Spain; 32Department of Endocrinology and Nutrition, Instituto de Investigación Sanitaria Hospital Clínico San Carlos (IdISSC), 28040 Madrid, Spain; pilar.matia@gmail.com; 33CIBER Diabetes y Enfermedades Metabólicas (CIBERDEM), Instituto de Salud Carlos III (ISCIII), 28029 Madrid, Spain; jovidal@clinic.cat; 34Department of Endocrinology, Institut d’Investigacions Biomédiques August Pi Sunyer (IDIBAPS), Hospital Clinic, University of Barcelona, 08036 Barcelona, Spain; 35Department of Endocrinology and Nutrition, Hospital Fundación Jimenez Díaz, Instituto de Investigaciones Biomédicas IISFJD, University Autonoma, 28040 Madrid, Spain; 36Department of Endocrinology and Nutrition, Hospital Clínic, 08036 Barcelona, Spain; SERRAMIR@clinic.cat; 37Unidad de Gestión Clínica Arroyo de la Miel, Distrito de Atención Primaria Costa del Sol, Servicio Andaluz de Salud, 29630 Benalmádena, Spain; rosa.carabano.sspa@juntadeandalucia.es; 38Osasunbidea, Servicio Navarro de Salud, Atención Primaria, 31003 Pamplona, Spain; 39IDIAP Jordi Gol i Gurina, 43202 Reus, Spain

**Keywords:** bioactive compounds, phytochemicals, dietary fats, Mediterranean diet, PREDIMED-Plus study, plasma carotenoids, matrix effect absorption, liquid chromatography, mass spectrometry

## Abstract

Carotenoids are pigments contained mainly in fruit and vegetables (F&V) that have beneficial effects on cardiometabolic health. Due to their lipophilic nature, co-ingestion of fat appears to increase their bioavailability via facilitating transfer to the aqueous micellar phase during digestion. However, the extent to which high fat intake may contribute to increased carotenoid plasma concentrations is still unclear. The objective was to examine the degree to which the consumption of different amounts of both carotenoid-rich foods and fats is associated with plasma carotenoid concentrations within a Mediterranean lifestyle context (subsample from the PREDIMED-Plus study baseline) where consumption of F&V and fat is high. The study population was categorized into four groups according to their self-reported consumption of F&V and fat. Carotenoids were extracted from plasma samples and analyzed by HPLC-UV-VIS-QqQ-MS/MS. Carotenoid systemic concentrations were greater in high consumers of F&V than in low consumers of these foods (+3.04 μmol/L (95% CI: 0.90, 5.17), *p*-value = 0.005), but circulating concentrations seemed to decrease when total fat intake was very high (−2.69 μmol/L (−5.54; 0.16), *p*-value = 0.064). High consumption of F&V is associated with greater systemic levels of total carotenoids, in particular when fat intake is low-to-moderate rather than very high.

## 1. Introduction

A high intake of phytochemicals from fruits and vegetables (F&V) has been linked to better cardiovascular health [1]. Among them, carotenoids, phytochemicals that give these foods their yellow to reddish shades [2], have been associated with decreased risk of type 2 diabetes [3], cardiovascular disease [4,5], and cancer [6,7]. Some carotenoids can be enzymatically converted into essential vitamin A, but the main mechanism that explains their salutary health effects seems to be their antioxidant action [8].

Carotenoids are very hydrophobic molecules which contain a long carbon chain rich in conjugated double bonds and they are classified based on their chemical structure: carotenes only contain hydrocarbons, while xanthophylls are oxygenated carotenes. Their specific molecular structure and physicochemical characteristics explain their storage in vegetable chromoplasts and conjugation with proteins. Therefore, they are strongly linked to the food matrix and, consequently, have low bioaccessibility. However, their absorption can be increased by simple processing and cooking methods [9,10,11] and, because of their lipophilic profile, by the use of oils during cooking [12]. Dietary fat appears to increase their bioaccessibility and bioavailabity and, hence, their plasma concentrations via the emulsification and facilitation of incorporation into mixed micelles during digestion [12,13]. Once liberated from the matrix, carotenoids are absorbed and distributed in the human body in a similar way to other dietary fat-soluble compounds. In this matter, genetic variability in cleavage, transport and metabolism proteins can also affect carotenoid plasma concentrations [14].

A minimum fat consumption of 3–5 g/day guarantees sufficient carotenoid absorption [9], and some reports have pointed out that greater fat intakes may increase their bioavailability [15,16]. Interestingly, no large studies have investigated whether this association is linear or if it plateaus at some point.

The objective of this study was to assess the association of F&V consumption and fat intake, individually and combined, with plasma concentrations of carotenoids in an older Mediterranean population with metabolic syndrome. The Mediterranean diet is one of the most renowned healthy dietary patterns, and it is rich not only in F&V, but also in healthy fats. This is a novel approach given that, to our knowledge, no studies have contemplated actual dietary intake in order to assess this relationship.

## 2. Materials and Methods

### 2.1. Study Design

This work has been carried out as a baseline cross-sectional analysis within the PREDIMED-Plus study, a 6-year, multicenter, randomized, parallel-group lifestyle intervention trial for the primary prevention of cardiovascular disease that involves 6874 participants recruited in 23 Spanish centers from September 2013 to November 2016 [17]. Eligible participants were overweight or obese (body mass index between 27 and 40 kg/m^2^) men and women ranging in age from 55 and 60 years, respectively, to 75 years. They all met at least three metabolic syndrome criteria according to the International Diabetes Federation and the American Heart Association and National Heart, Lung, and Blood Institute [18]. The selection and the description of the study sample have been reported elsewhere [19]. Details on the study protocol can be found at http://www.predimedplus.com/ (accessed on 16 October 2020).

#### 2.1.1. Ethics Statement

The study was conducted according to the ethical standards’ guidelines of the Declaration of Helsinki and all procedures involving human participants and patients were approved by the Institutional Review Boards of the participating centers. The clinical trial was registered in the ISRCTN of London, England with the number 89898870 on 24 July 2014. Written informed consent was obtained from all participants.

#### 2.1.2. Sample Selection

Dietary intake was determined using the validated, semi-quantitative 143-item PREDIMED-Plus food frequency questionnaire of the year prior to inclusion [19]. According to the consumption of F&V and total dietary fat, we randomly selected four groups of 60 participants similar in age, sex, anthropometric measurements and cardiovascular risk factors, who fulfilled the following characteristics: (1) participants with low F&V consumption (first decile) and low-to-moderate fat (first quartile) intake (reference); (2) participants with low F&V consumption (first decile) and very high fat intake (fourth quartile); (3) participants with high F&V consumption (tenth decile) and low-to-moderate fat intake (first quartile); and (4) participants with high F&V consumption (tenth decile) and very high fat intake (fourth quartile).

#### 2.1.3. Covariates

Trained personnel collected baseline data on: age; sex; prevalence of diabetes, hypercholesterolemia, and hypertension; body mass index; physical activity; and smoking habit as previously described [19]. From the validated food frequency questionnaire, we also estimated the consumption in g/day of alcohol and energy intake in kcal/day. Finally, based on the 17-item energy- reduced Mediterranean diet adherence score [20], we estimated the overall quality of diet regardless of the consumption of fruits, vegetables, dietary fats and alcohol (this modified version was obtained using the 9 questions that were independent from the consumption of these items; the higher the value of the score obtained, the higher the overall quality of the diet).

#### 2.1.4. Sample Size Calculation

A sample size of 60 participants per group allowed ≥80% power to detect a significant difference of 0.42 μg/mL in the concentration of total carotenoids in plasma between groups, considering a 2-sided type I error of 0.05, a loss rate of 5%, and the standard deviation of the differences in plasma concentration of total carotenoids in middle-aged Spanish adults (SD = 0.80) [21].

### 2.2. Carotenoids Extraction and Analysis

#### 2.2.1. Standards and Samples

K3-EDTA fasting plasma samples from the baseline blood extractions were analyzed. These samples were drawn in the first visit of the study, just after being randomly assigned to an intervention group and stored at −80 °C until use. All samples and standards were always handled avoiding exposure to light and under cool conditions. Carotenoid standards: astaxanthin, canthaxanthin, *E*-*β*-apo-8′-carotenal, *α*-carotene, *β*-carotene, fucoxanthin, and lycopene were purchased from Sigma-Aldrich (St. Louis, MO, USA). Lutein was provided by Cayman Chemical (Ann Arbor, MI, USA), zeaxanthin and *β*-cryptoxanthin were purchased from Extrasynthese (Genay, Lyon, France). 13-*Z-β*-carotene and 9-*Z-β*-carotene were purchased from Carbosynth (Newbury, Berkshire, UK). Standards were stored in powder form at −20 °C and protected from light.

Methanol of LC-MS grade, n-hexane, ethanol and methyl tert-butyl ether (MTBE) of HPLC grade, blank human plasma and butylated hydroxytoluene (BHT) were obtained from Sigma-Aldrich. Ammonium acetate (AMAC) and acetic acid of HPLC grade were purchased from Panreac Quimica SLU (Barcelona, Spain). Ultrapure water (Milli-Q) was generated by a Millipore system (Bedford, MA, USA).

#### 2.2.2. Extraction

In order to avoid oxidation and isomerizing, the samples were extracted and analyzed in a room with filtered light and kept in ice at all times. The extraction was performed using a method previously developed by our group [22]. Briefly, 450 μL of the samples were thawed and mixed with 800 μL of ethanol, 500 μL of ultrapure water and 2 mL of n-hexane/BHT (100 mg/L) in crystal tubes and vortexed for 1 min. 100 μL of fucoxanthin at 1 mg/mL were also added as internal standard. Then, they were centrifuged at 2070× *g* for 5 min at 4 °C and the upper nonpolar layer was separated into a new tube. The lower aqueous phase underwent re-extraction with 2 more milliliters of n-hexane/BHT (100 mg/L), 1 min of vortexing and 5 min of centrifugation at 2070× *g* and 4 °C. The upper nonpolar layer was again separated and combined with the first one to undergo evaporation to dryness by a sample concentrator under nitrogen gas at room temperature. The evaporate was then reconstituted with 100 μL of methanol and stored in glass amber vials with inserts at −80 °C until the day of analysis.

The same procedure was followed to prepare calibration curves. Stock solution of blank human plasma was used to this end.

#### 2.2.3. HPLC-UV-VIS-MS/MS Analysis

After extraction, the carotenoids were analyzed using a YMC Carotenoid S-5 μm, 250 × 4.6 mm (Waters, Milford, MA, USA) column for separation coupled for detection to a UV-VIS detector set at 450 nm and a triple quadrupole mass spectrometer QTRAP4000 (Sciex, Foster City, CA, USA) equipped with APCI ionization source and controlled by Analyst v.1.6.2 software (Sciex). The column was maintained at 40 °C throughout the analysis [23].

The chromatographic separation was achieved by means of the combination of two mobile phases. Mobile phase A consisting of methanol, AMAC at 0.7 g/L and 0.1% of acetic acid. Mobile phase B consisting of MTBE and methanol (80:20, *v*/*v*), AMAC at 0.7 g/L and 0.1% of acetic acid. The mobile phase A gradient conditions used were (t (min),%): (0.0, 90); (10.0, 75); (20.0, 50); (25.0, 30); (35.0, 10); (37.0, 6); (39.0, 90); (50.0, 90). The flow rate was 0.6 mL/min and total run time of analysis 50 min. 20 μL of the sample were injected into the system.

Quantitation was achieved by the construction of calibration curves for each compound and interpolation into them using MultiQuant software version 3.0.1 (Sciex) by the internal standard method. Due to the labile profile of the *Z*-lycopene standard, this carotenoid was quantified in *E*-lycopene equivalents. Detection and quantification limits, concentration ranges, and correlation coefficients of the calibration curves prepared in blank human plasma for the eleven compounds are shown in Appendix A.

### 2.3. Statistical Analysis

Baseline characteristics of the participants are presented as means ± standard deviations for continuous variables and percentages for categorical variables. To determine possible differences in baseline characteristics between groups, we used one-way ANOVA for continuous variables and χ^2^-test for categorical variables. Nine samples were excluded from the analyses for they had implausible energy intakes reported (>3500 Kcal/day for women and >4000 Kcal/day for men) [24].

Considering the particular nature of carotenoids as a biomarker (we were unable to quantify some sub-species in some participants since they were below the limit of quantification), we studied the inter-group differences using adapted survival regression models as described by Helsel DR [25] using the ‘survival’ package in R Software version 4.0.0. We first determined whether there was a significant interaction between the groups according to F&V and fat intake and carotenoid levels. We adapted survival regression models where the plasma carotenoid concentration was the dependent variable and applied a likelihood ratio test between the nested models with and without an interaction product-term of “F&V intake group × fat intake group”. Any *p*-value < 0.1 for the interaction was considered as significant following the strategy described by other authors [26,27]. We used three regression models of increasing complexity. Model 1 was adjusted for age (continuous) and sex. Model 2 was further adjusted for physical activity (continuous). Model 3 was additionally adjusted for total energy intake, the modified Mediterranean diet adherence score and alcohol consumption (continuous, all).

Statistical analyses were performed for individual carotenoids and also for the two groups of carotenoids: carotenes (*α*-carotene, *β*-carotene, *E*-lycopene and *Z*-lycopene) and xanthophylls (astaxanthin, lutein, canthaxanthin and *β*-cryptoxanthin), for the sum of lycopenes and for the total sum of total measured carotenoids. We used standard statistical methods from the ‘survival’ package. The differences between groups are expressed as median changes (95% confidence intervals, CI). *p*-values < 0.05 were deemed to be significant.

## 3. Results

### 3.1. Participant Characteristics

All four groups, comprising 106 women and 124 men (one sample was unavailable for analysis), were comparable in terms of age, sex and cardiovascular risk factors (Table 1). Around 70% of participants were obese, and the remaining were overweight, overall the four groups were comparable in terms of body mass index. Their burden of cardiovascular risk factors was high: 87% had hypertension, 67% hypercholesterolemia, and 23% diabetes. Only slightly over 15% of the participants were smokers. Those who consumed more F&V tended to be more physically active. The participants in the first decile of F&V consumption had an average consumption of 289 g/day of F&V and those in the first quartile of fat intake had an average intake of fat of 67 g/day. At the other end, intakes in the highest quantiles were both very high, 1295 g of F&V/day and 141 g of fat/day. Inter-group differences in food and nutrient intake are detailed in Appendix A.

### 3.2. Carotenoid Concentration in Plasma

Astaxanthin, lutein, canthaxanthin, *β*-cryptoxanthin, *α*-carotene, *β*-carotene and lycopene were the predominant carotenoids found in plasma, as described in other studies [28,29,30]. The determination of other carotenoids was below the limit of quantification.

#### 3.2.1. High F&V vs. Low F&V

Relative to low F&V consumption (reference according to the model after adjustment for several covariates), individuals with high F&V consumption showed greater plasma total carotenoid concentrations (+3.04 μmol/L (95% CI: 0.90, 5.17), *p*-value = 0.005). The differences attained statistical significance when the intake of fat was low-to-moderate (+3.83 μmol/L (0.97, 6.7), *p*-value = 0.009, *P* for interaction = 0.161), but not when the intake of fat was very high. Likewise, the plasma concentration of carotenes and xanthophylls significantly increased (+2.80 μmol/L (0.46; 5.14) and +0.88 μmol/L (0.48; 1.27), *p*-values = 0.019 and <0.001, *P* for interaction = 0.267 and 0.073, respectively) particularly when fat intake was low-to-moderate (+3.53 μmol/L (0.38; 6.68) and +1.00 μmol/L (0.48; 1.53), *p*-values = 0.028 and < 0.001, respectively), but not when it was very high (Table 2).

#### 3.2.2. Very High Fat vs. Low-to-Moderate Fat Intake

In relation to low-to-moderate fat intake, participants with very high fat consumption tended to present with a lower concentration of carotenoids (−2.69 μmol/L (−5.54; 0.16), *p*-value = 0.064), particularly, although the difference did not achieve statistical significance, when the consumption of F&V was high (−2.52 μmol/L (−6.10; 1.05), *p*-value = 0.166) (Table 3). When the consumption of F&V was low, no differences were observed between the low-to-moderate fat intake and the very high fat intake groups. Likewise, the plasma concentration of carotenes tended to decrease, and the concentrations of xanthophylls significantly decreased (−2.36 μmol/L (−5.51; 0.79) and −0.88 μmol/L (−1.41; −0.35), *p*-values = 0.142 and 0.001, respectively), particularly, although not significantly for carotenes, when F&V intake was high (−2.05 μmol/L (−5.97; 1.87) and −0.78 μmol/L (−1.44; −0.13), *p*-values = 0.305 and 0.019, respectively).

#### 3.2.3. Comparisons between Extreme Values

When comparing the groups with extreme conditions (high F&V + low-to-moderate fat intake vs. low F&V + very high fat intake), the inter-group difference achieved statistical significance (+3.86 μmol/L (0.86; 6.85), *p*-value = 0.012). Xanthophyll differences were less dispersed but smaller in magnitude (+1.20 μmol/L (0.65; 1.75), *p*-value < 0.001) relative to carotenes (+3.40 (0.079; 6.72), *p*-value = 0.045). No significant differences were found between the group high in both F&V and fat and the group low in F&V and low-to-moderate in fat (Table 4).

Appendix A are extended versions of the results tables, showing the comparisons for individual carotenoids.

## 4. Discussion

High consumption of F&V is associated with greater concentration of total carotenoids in plasma, especially when dietary fat intake is low-to-moderate (below 70 g/day), but, unexpectedly, not when fat intake is very high (over 140 g/day). These associations are particularly striking for xanthophylls but of greater magnitude for carotenes. To the best of our knowledge, this is the first analysis highlighting the relevance of both the source (F&V) and the vehicle (fat) involved in carotenoid bioavailability using high-throughput metabolomic biomarkers in a large cohort study.

The Mediterranean diet is characterized by a high consumption of F&V along with healthy fats, mainly olive oil and nuts. The mean consumption of F&V at baseline in the PREDIMED-Plus study participants, a Mediterranean population following unrestrained diets, was 685.4 g/day. This is higher than the average of 345 g F&V /day reported for the Spanish population in the ANIBES (Anthropometry, Intake and Energy Balance in Spain) study [31,32], and also higher than the average intake reported for Europe [33]. The WHO recommends an intake of at least 600 g/day of fruits and vegetables in order to prevent diseases and micronutrient deficiencies [34]. In this study, the group comprising participants from the first decile of F&V consumption did not meet the minimum recommended amount, while the tenth decile complied with it by a large margin.

Mean fat intake in the baseline analysis of the PREDIMED-Plus study was 105.3 g/day, which represents 39.4% of daily energy intake, although saturated fat accounted for less than 10%. Dietary Guidelines for Americans 2020–2025 do not specify an upper limit for total fat intake [35]. Overall, around 70 g of fat per day has been defined as a regular amount [36]. However, fat is a crucial nutrient in the Mediterranean diet, which explains why mean dietary fat intake in the first decile was only slightly below 70 g/day, a value equivalent to approximately 30% of daily energy intake.

In individual participant meta-analysis of controlled feeding studies, robust evidence was presented for a positive dose-response association between F&V consumption and total plasma carotenoid concentrations [37]. Indeed, in this study total carotenoids and also carotenes and xanthophylls were in higher concentration in the group that reported high F&V consumption. Surprisingly, when separating the groups according to dietary fat intake, the groups with very high fat intake did not show significant differences in carotenoid plasma concentrations compared to the low-to-moderate fat intake group. This suggests there may be a limit up to which a positive association between higher fat intake with higher carotenoid concentrations in plasma is observed, as the group reporting very high fat and high F&V consumption did not have the highest systemic carotenoid concentrations.

The findings related to the comparison between high and low-to-moderate fat intake groups in this study are also thought-provoking: total carotenoids tended to be in higher concentration in the low-to-moderate fat intake group. Similar intriguing results were obtained when separating the groups with the same F&V consumption: there were no differences in plasma carotenoid concentration between low F&V consumption groups, and total carotenoids tended to be, although not significantly, in lower concentration in the high F&V and fat consumption group. Even though the comparison between the completely opposite groups is not so typical, it gives hints of a similar outcome: the group with high F&V and fat consumption did not show sizable differences compared to the group with low F&V and low-to-moderate fat consumption. If fat was positively associated with plasma carotenoid concentration regardless of their concentration, these two groups would have likely been significantly different. The group with high F&V consumption but low-to-moderate in fat intake disclosed the expected results, significantly higher carotenoid concentrations in comparison to the group with low F&V consumption and very high fat intake, which makes sense, as even if a high intake of fat was associated with higher carotenoid concentrations, the group with low consumption of F&V may not have had the carotenoid content available to begin with to reach the levels observed with a high F&V intake, for they are in extreme deciles.

In an intervention study with a similar sample size (*n* = 122), Djuric et al. (2004) found that the combination of a low fat and high F&V intervention, but not a F&V intervention alone (without fat intake recommendations), significantly increased the plasma concentrations of xanthophylls, indicating an interaction between F&V and fat in, at least, xanthophyll absorption [38]. The small sample size of both Djuric’s and this study limit the testing of the interaction for total carotenoids, as carotenes are in higher concentrations and the modification of the effect is more difficult to detect.

These results taken together lead to the hypothesis of a dual role of dietary fat in carotenoid appearance in human plasma: when scarce carotenoids are available, even moderate fat intake is sufficient to escort them to be absorbed. However, when consuming high amounts of F&V, very high fat intake might not further increase carotenoid bioavailability. This might be explained by the absorption mechanisms of these phytochemicals: carotenoids must be released from the food matrix, emulsified into oil droplets and further integrated into mixed micelles in order to be absorbed at the brush border of the enterocytes, which can take place by either passive diffusion or active uptake via fat transporters [13,39]. Fat in the intestinal lumen would promote carotenoid bioaccessibility by enhancing their extractability and micellarization but, when in very high concentrations, it could hinder bioavailability by saturation of mixed micelles [40] or competition for the transporters. Bile acids also take an active part in the absorption process of carotenoids and, because their secretion can be altered by dietary fat intake [41], they could also be influencing the final carotenoid concentration in plasma [14]. Altogether, these results suggest that, even though carotenoids perform as biomarkers when assessing F&V consumption, the overall dietary components must be carefully considered.

The main limitation of this study is the impossibility to determine food processing from the food frequency questionnaires, along with its cross-sectional design. These factors preclude establishing a firm conclusion on the association between fat consumption and carotenoid absorption. In addition, the absence of publications evaluating these amounts of fat and carotenoids did not allow comparison of the results with previous studies in similar populations.

Arranz et al. (2015) showed that addition of 10% of olive oil to tomato juice improved lycopene isomerization and this was associated with reduction of total and LDL-cholesterol values compared to tomato juice alone [42]. However, this and all studies reviewed that compared carotenoid absorption in relation to different types of fat have been conducted using only modest amounts of fat (2.5% to 10%) [43], and most of them were performed in animal models [44] or cell cultures [40]. Interestingly, the latter stated that carotenoid micellarization did not increase when higher amounts of fat were used, suggesting that there was micellar saturation, a finding that might help explain our results. In this way, it would be interesting to study whether a regular diet rich in F&V as well as healthy fats (such as extra virgin olive oil) could lead to higher plasma carotenoid concentrations and, in this way, enhance their salutary biological effects.

## 5. Conclusions

Differences in plasma carotenoids concentration depending on the amount of fat consumed were determined in a sub-sample of participants from the PREDIMED-Plus study at baseline, concerning adults in real-life conditions. Higher plasma carotenoid concentrations were found when higher amounts of F&V were consumed, but very high dietary fat intake was associated with lower carotenoid concentrations in plasma when compared to low-to-moderate fat intake. Clinical trials using different amounts of fat for outcomes of plasma biomarkers of F&V consumption are warranted to eventually confirm our results and establish whether the dual role of dietary fat in carotenoid absorption needs to be controlled for optimizing nutrition. Further research is needed to confirm these results and determine what the critical point is for fat to facilitate carotenoid absorption. Further analyses could also examine whether different types of fat, even in high quantities, are associated with different plasma carotenoid concentrations.

## Figures and Tables

**Table 1 antioxidants-10-00473-t001:** Participant characteristics by Fruit and Vegetable and Fat Intake groups.

Characteristics	All	Low F&V		High F&V		*p*-Value *
		Low-to- Moderate Fat	Very High Fat	Low-to- Moderate Fat	Very High Fat	
No. of subjects	230	59	58	60	53	
Age, years	66.1 ± 4.40	65.9 ± 4.46	66.3 ± 3.61	65.8 ± 5.07	66.2 ± 4.25	0.935
Women, *n* (%)	106 (46.1)	26 (44.1)	26 (44.8)	32 (53.3)	22 (41.5)	0.604
Type-2 diabetes mellitus, *n* (%)	55 (23.6)	11 (18.6)	11 (19.0)	21 (35.0)	11 (20.8)	0.135
Hypercholesterolemia, *n* (%)	155 (67.4)	39 (66.1)	42 (72.4)	40 (66.7)	34 (64.2)	0.811
Hypertension, *n* (%)	200 (87.0)	50 (84.7)	53 (91.4)	53 (88.3)	44 (83.0)	0.510
Body mass index, kg/m^2^	32.7 ± 3.50	32.0 ± 2.99	32.7 ± 3.64	32.4 ± 3.76	33.6 ± 3.53	0.100
Current smoker, *n* (%)	37 (16.1)	8 (13.6)	10 (17.2)	6 (10.0)	13 (24.5)	0.189
Leisure-time physical activity, MET·min/week	2525 ± 2458	1780 ± 1855	2276 ± 1890	3064 ± 3248	3019 ± 2368	0.011

F&V, fruit and vegetables; MET, metabolic task equivalents. Values are percentages for categorical variables and means ± SD for continuous variables. * *p*-values were calculated by analysis of variance–one factor was used for continuous variables and the χ^2^-test for categorical variables, *p* < 0.05.

**Table 2 antioxidants-10-00473-t002:** Differences in total carotenoid, carotenes and xanthophylls plasma concentrations (μmol/L) between F&V consumption groups.

		High F&V vs. Low F&V	*p*-Value	High F&V vs. Low F&V (Low-to- Moderate Fat)	*p*-Value	High F&V vs. Low F&V (High Fat)	*p*-Value
Total carotenoids	Median	5.31 vs. 2.08		6.75 vs. 2.48		4.23 vs. 1.71	
	ß [CI]-model 1	3.64 [1.85; 5.44]	<0.001	5.01 [2.54; 7.48]	<0.001	2.12 [−0.43; 4.67]	0.104
	ß [CI]-model 2	2.97 [1.18; 4.76]	0.001	4.13 [1.67; 6.60]	0.001	1.67 [−0.84; 4.18]	0.192
	ß [CI]-model 3	3.04 [0.90; 5.17]	0.005	3.83 [0.97; 6.70]	0.009	1.33 [−1.64; 4.30]	0.379
Carotenes	Median	3.00 vs. 0.25		4.26 vs. 0.95		1.36 vs. 0.19	
	ß [CI]-model 1	3.47 [1.50; 5.44]	<0.001	4.70 [1.99; 7.42]	<0.001	2.06 [−0.75; 4.88]	0.150
	ß [CI]-model 2	2.77 [0.79; 4.74]	0.006	3.79 [1.07; 6.51]	0.006	1.60 [−1.17; 4.37]	0.257
	ß [CI]-model 3	2.80 [0.46; 5.14]	0.019	3.53 [0.38; 6.68]	0.028	1.35 [−1.92; 4.62]	0.419
Xanthophylls	Median	2.04 vs. 1.09		2.44 vs. 1.04		2.03 vs. 1.09	
	ß [CI]-model 1	1.00 [0.67; 1.33]	<0.00001	1.27 [0.81; 1.73]	<0.00001	0.69 [0.22; 1.17]	0.004
	ß [CI]-model 2	0.89 [0.55; 1.23]	<0.00001	1.13 [0.67; 1.59]	<0.00001	0.62 [0.15; 1.09]	0.009
	ß [CI]-model 3	0.88 [0.48; 1.27]	<0.001	1.00 [0.48; 1.53]	<0.001	0.42 [−0.13; 0.96]	0.136

ß, difference between groups; CI, confidence interval. Model 1—adjusted for age and sex. Model 2—adjusted for age, sex and physical activity. Model 3—adjusted for the variables used in model 2 plus energy intake, the modified Mediterranean diet adherence score (subtracting the questions regarding F&V, fat and wine) and alcohol consumption (g/day). *p*-values < 0.05 were considered significant.

**Table 3 antioxidants-10-00473-t003:** Differences in total carotenoid, carotenes and xanthophylls plasma concentrations (μmol/L) between fat intake groups.

		High fat vs. Low-to-Moderate Fat	*p*-Value	High Fat vs. Low-to-Moderate Fat (Low F&V)	*p*-Value	High Fat vs. Low-to-Moderate Fat (High F&V)	*p*-Value
Total carotenoids	Median	2.35 vs. 5.02		1.71 vs. 2.48		4.23 vs. 6.75	
	ß [CI]-model 1	−1.46 [−3.31; 0.39]	0.122	0.039 [−2.45; 2.53]	0.976	−2.85 [−5.39; −0.30]	0.028
	ß [CI]-model 2	−1.59 [−3.37; 0.19]	0.081	−0.29 [−2.72; 2.15]	0.816	−2.75 [−5.24; −0.27]	0.030
	ß [CI]-model 3	−2.69 [−5.54; 0.16]	0.064	−0.021 [−3.54; 3.50]	0.991	−2.52 [−6.10; 1.05]	0.166
Carotenes	Median	0.37 vs. 2.48		0.19 vs. 0.95		1.36 vs. 4.26	
	ß [CI]-model 1	−1.19 [−3.21; 0.82]	0.245	0.23 [−2.55; 3.00]	0.873	−2.41 [−5.18; 0.35]	0.087
	ß [CI]-model 2	−1.32 [−3.27; 0.63]	0.185	−0.12 [−2.85; 2.60]	0.928	−2.31 [−5.02; 0.39]	0.094
	ß [CI]-model 3	−2.36 [−5.51; 0.79]	0.142	0.13 [−3.81; 4.07]	0.948	−2.05 [−5.97; 1.87]	0.305
Xanthophylls	Median	1.25 vs. 1.54		1.09 vs. 1.04		2.03 vs. 2.44	
	ß [CI]-model 1	−0.35 [−0.71; 0.002]	0.051	−0.045 [−0.51; 0.42]	0.847	−0.62 [−1.09; −15]	0.010
	ß [CI]-model 2	-0.38 [−0.72; −0.034]	0.031	−0.10 [−0.55; 0.35]	0.668	−0.61 [−1.07; −0.14]	0.010
	ß [CI]-model 3	−0.88 [−1.41; −0.35]	0.001	−0.19 [−0.84; 0.45]	0.555	−0.78 [−1.44; −0.13]	0.019

ß, difference between groups; CI, confidence interval. Model 1—adjusted for age and sex. Model 2—adjusted for age, sex and physical activity. Model 3—adjusted for the variables used in model 2 plus energy intake, the modified Mediterranean diet adherence score (subtracting the questions regarding F&V, fat and wine) and alcohol consumption (g/day). *p*-values < 0.05 were considered significant.

**Table 4 antioxidants-10-00473-t004:** Differences in total carotenoid, carotenes and xanthophylls plasma concentrations (μmol/L) between extreme groups.

		Low-to-Moderate Fat & High F&V vs. High Fat & Low F&V	*p*-Value	High Fat & High F&V vs. Low-to-Moderate Fat & Low F&V	*p*-Value
Total carotenoids	Median	6.75 vs. 1.71		4.23 vs. 2.48	
	ß [CI]-model 1	4.97 [2.49; 7.45]	<0.001	2.16 [−0.39; 4.70]	0.096
	ß [CI]-model 2	4.42 [1.98; 6.86]	<0.001	1.38 [−1.15; 3.91]	0.284
	ß [CI]-model 3	3.86 [0.86; 6.85]	0.012	1.31 [−3.27; 5.89]	0.575
Carotenes	Median	4.26 vs. 0.19		1.36 vs. 0.95	
	ß [CI]-model 1	4.48 [1.75; 7.21]	0.001	2.29 [−0.52; 5.10]	0.110
	ß [CI]-model 2	3.92 [1.22; 6.61]	0.004	1.48 [−1.32; 4.27]	0.300
	ß [CI]-model 3	3.40 [0.079; 6.72]	0.045	1.48 [−3.58; 6.54]	0.567
Xanthophylls	Median	2.44 vs. 1.09		2.03 vs. 1.04	
	ß [CI]-model 1	1.32 [0.86; 1.77]	<0.00001	0.65 [0.18; 1.12]	0.007
	ß [CI]-model 2	1.23 [0.77; 1.68]	<0.00001	0.52 [0.049; 0.99]	0.031
	ß [CI]-model 3	1.20 [0.65; 1.75]	<0.001	0.22 [−0.62; 1.06]	0.608

ß, difference between groups; CI, confidence interval. Model 1—adjusted for age and sex. Model 2—adjusted for age, sex and physical activity. Model 3—adjusted for the variables used in model 2 plus energy intake, the modified Mediterranean diet adherence score (subtracting the questions regarding F&V, fat and wine) and alcohol consumption (g/day). *p*-values < 0.05 were considered significant.

## Data Availability

There are restrictions on the availability of data for the PREDIMED-Plus trial, due to the signed consent agreements around data sharing, which only allow access to external researchers for studies following the project purposes. Requestors wishing to access the PREDIMED-Plus trial data used in this study can make a request to the PREDIMED-Plus trial Steering Committee chair: jordi.salas@urv.cat. The request will then be passed to members of the PREDIMED-Plus Steering Committee for deliberation.

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
