# Peer review of "High Fruit and Vegetable Consumption and Moderate Fat Intake Are Associated with Higher Carotenoid Concentration in Human Plasma"

_antioxidants, 2021, doi:10.3390/antiox10030473_

Round 1

Reviewer 1 Report

This is an interesting Ms for all of those interested in MD, mainly from a practical point of view: the importance of cooking. We all say that cooking carotene rich vegetables with olive oil is the best way to get high availability of carotenes. That does not mean that the diet has to be a high fat diet. The Ms shows it without any doubt. But it lacks some informations: which kind of fat, which kind of recipes . You recognize that you did not have this kind of information  , but I am sure that the PREDIMED questionnaire  allows for a specific identification of the fat intake. It could be intersting to introduce this specificity in the high and low fat groups.

a minor point: in the introduction you say that consumption of phytochemicals from fruit and vegetables participate i the protection against diabetes type2 and CVD. Cancers should also be mentionned.

Reviewer 2 Report

Thank you for doing research in such needed area. We need more research like this looking into dietary intake and its relation to our health. English language is very good. No changes needed. Not sure why the only good option we are given is stating "minor spell check required", the reason I had to select it, as it was the best option available.

Author Response

Thank you very much for such kind comments!

Reviewer 3 Report

The current manuscript investigates the association of F&V consumption and fat intake, with plasma concentrations of carotenoids in an older Mediterranean population with metabolic syndrome. It is an interesting investigation, although I do have some remarks:

Introduction:

From the introduction it is not initially clear that the population are (older) people with metabolic syndrome. What is already known? And how can this population be compared with the ‘normal’ population as they have lots of other problems (they not only differ in higher fat intake, but there might also be multiple confounders).

Study design:

From the sample size it seems that 4 times 60 participants per group (240 in total) will be included. However, from the study results it seems that in total 230 participants were included. Are the 9 + 1 excluded samples from the analyses the remaining 10 cases?

Study population:

People with quite a high BMI (27-40) were included in this study, however BMI is not taken into account as covariate in the analyses. Could there be an effect of BMI as well? Did the authors investigated this in a model, like the Model 1/2/3 with the current adjustments made?

Results:

From the baseline table (Table 1) it seems that Type-2 DM is, although not significantly different, quite high in the High F&V + low-to-moderate fat group compared to the other groups. Did the authors investigate this further? Could this be of influence on the current presented results?

Astaxanthin, lutein, canthaxanthin, B-cryptoxanthin, a-carotene, B-carotene and lycopene were the predominant carotenoids found in plasma. Are these results also conform other studies? Or likely that these were most predominant?

Discussion:

The authors state that the mean F&V consumption is higher as well as the mean fat intake compared to the average intake / regular reported amount. One of the inclusion criteria was a(n) (extremely) high BMI, suggesting the intake of food consumption would be logically quite high compared to people with a more common / normal BMI. In which way could this be of influence? And how should this be discusses/results interpreted.

Reviewer 4 Report

The authors showed the interplay between fat and carotenoids in human. The manuscript provides an information on the uptake of carotenoid depend on the amount of fats. However it lacks a little bit of novelty because the interaction between high/low fat diet and the plasma concentration of carotenoid have been reported before (Roodenburg AJ, Leenen R, van het Hof KH, Weststrate JA, Tijburg LB. Amount of fat in the diet affects bioavailability of lutein esters but not of alpha-carotene, beta-carotene, and vitamin E in humans. Am J Clin Nutr. 2000 May;71(5):1187-93.). In addition, the effects of lipid on the bioaccessibility of carotenoids in the spinach have been investigated (Akihiko NAGAO, Bioavailability of Dietary Carotenoids: Intestinal Absorption and Metabolism, Japan Agricultural Research Quarterly: JARQ, 2014, Volume 48, Issue 4, Pages 385-391). In this paper, trioleoylglycerol and oleic acid could reduce the bioaccessibility of lutein. How about the composition of fat diet used in the manuscript? The fat composition is important to explain why circulating concentrations seemed to decrease when total fat intake was very high. Main composition of olive oil is oleic acid. So oleic acid may contribute to the decrease in uptake of carotenoid in the high fat condition. Please compare to previous reports and add discussion or mention difference from those reports.

Did you investigate the concentration of bile acids in plasma? because bile acid is essential for lipid absorption in the gut and related to an uptake of carotenoids (El-Gorab MI, Underwood BA, Loerch JD. The roles of bile salts in the uptake of beta-carotene and retinol by rat everted gut sacs. Biochim Biophys Acta. 1975 Aug 20;401(2):265-77.). The bile acids concentration in plasma will confer a novelty on this manuscript. 

Round 2

Reviewer 4 Report

Please add an information of the interaction between bile induced by fat diet and carotenoid absorption to Discussion section. Because fat diet can cause change in bile acids composition associated with carotenoid absorption (doi: 10.1021/acs.jafc.9b00249; doi: 10.1002/mnfr.201600685 ).

Author Response

Thank you for the remark, the following has been added to line 418: "Bile acids also take an active part in the absorption process of carotenoids and, because their secretion can be altered by dietary fat intake (doi: 10.1021/acs.jafc.9b00249), they could also be influencing the final carotenoid concentration in plasma (doi: 10.1002/mnfr.201600685)."